# Nurses’ Work-Related Mental Health in 2017 and 2020—A Comparative Follow-Up Study before and during the COVID-19 Pandemic

**DOI:** 10.3390/ijerph192315569

**Published:** 2022-11-23

**Authors:** Cicilia Nagel, Kerstin Nilsson

**Affiliations:** 1Division of Occupational and Environmental Medicine, Department of Laboratory Medicine, Lund University, 22184 Lund, Sweden; 2Division of Public Health, Kristianstad University, 29128 Kristianstad, Sweden

**Keywords:** work-related, mental-health diagnoses, work situation, work environment, nurses

## Abstract

The COVID-19 pandemic put a lot of strain on healthcare organizations. Nurses account for over 50% of healthcare staff, and how nurses perform in their work is influenced by a number of human and work environmental factors. However, to our knowledge, there has not been a previous study with the intention to look at all areas that affect a sustainable working life and how these impact nurses’ mental well-being. The aim of this study is to investigate the association between, and the effect of, different factors in nurses’ work situations associated with nurses’ work-related mental-health diagnoses, before and during the COVID-19 pandemic. A questionnaire was sent out to all 9219 nurses in the Swedish county of Skane in the spring of 2017 and during wave two of the COVID-19 pandemic in the fall of 2020. The data were analyzed through logistic regression analysis. The results showed that lack of joy in the daily work, an increased workload and lack of support from co-workers had an increased association with work-related mental-health diagnoses. Future research regarding the long-term impact of COVID-19 on all areas of nurses’ professional and personal lives is needed.

## 1. Introduction

Healthcare workers around the world, primarily the nurses, were on the frontline of the Coronavirus pandemic that started in 2019 [1]. The second wave of the pandemic in Sweden occurred during the fall/winter 2020, the most serious wave of the pandemic regarding the burden on the healthcare sector. The pandemic has been described as a gigantic strain experiment on healthcare organizations, especially on healthcare staff due to exposure to hazards, such as psychological distress, fatigue, and trauma [1,2,3,4,5]. Healthcare workers had to perform their duties and face higher risks to their own health, such as the risk of infection [1]. Many nurses were afraid to become ill and die of COVID-19, which was incredibly stressful [6].

Nurses comprise half of the global health workforce [7,8]; however, for much of the general public, it is not fully understood what nurses do. Being a nurse includes promoting health, preventing illness as well as caring for people who are ill, disabled and dying. Advocating patients’ rights, promoting a safe environment, conducting and/or taking part in research and education are also key nursing roles [9]. Nurses are often the first healthcare staff that the patient encounters. Their roles may vary depending on workplace, but often include triage, early recognition of life-threatening conditions, administration of medications, performance of life-saving procedures and initiation of early referral [7]. Despite nurses being a common profession in healthcare, there is an increasing demand for nurses worldwide [8]. According to the World Health Organization (WHO) [7], one out of six of the world’s nurses are expected to retire in the next 10 years. Several countries experienced a lack of healthcare workers prior to the pandemic and many nurses are considering leaving the profession [10]. There is a serious nursing shortage in most European countries, which is insufficient to meet current healthcare demands [11,12]. Due to an ageing population, healthcare demands are predicted to increase, resulting in an estimated worldwide shortage of almost 6 million nurses by 2035 [7]. The ageing segment of the population is rapidly expanding and, thus, consuming more health services. Unfortunately, fewer new nurses are entering the work force; therefore, providing a healthy work environment to retain nurses in their workplace is essential for sustaining the profession [13].

Being able to work has a significant positive impact on people’s health, and healthy workplaces are beneficial not only for employees but also for organizations and for society [14]. Decent work is one of the UN Global Goals for sustainable society development [15]. A sustainable work situation for employees is significant for a healthy organization that attracts people to work as well as promoting better health for employees, thus, also giving a better possibility for employability to an increased age [16,17,18,19]. Working in a hospital can be complicated due to the interaction between patients, nurses and the organization. This can, under normal circumstances, cause problematic work situations, but during the pandemic this was likely even more of a factor. It is, therefore, important to detect problems and shortcomings in the work situation in order to improve and support healthy and sustainable employability and understand what measures need to be taken. Areas of employability, and whether individuals can and want to work or not, has been stated as nine impact and determinant areas connected to sustainable healthy working life in the SwAge-model [6,16,17,18,19], i.e., (1) the employees self-rated health and diagnoses; (2) factors in the physical work environment; (3) factors in the mental work environment; (4) having time for recuperation within the employees’ working hours, breaks and work pace; (5) the personal financial situation; (6) the employees’ personal social environment outside of work; (7) the work social environment at the workplace, with leadership, colleagues, etcetera; (8) factors related to whether the employee experiences stimulation and motivation within work tasks and appreciation from the organization/work place regarding their performed tasks; (9) if the employees’ have the right competence, skills and possibility for knowledge development in work.

As earlier mentioned, nurses play a key role in delivering care to patients [11]. How the nurses perform in their work environment is influenced by a number of human and environmental factors, including the type of information available, work experience, ambiguity, unpredictability, conflicting goals and time pressure [20]. Nurses face a higher risk of developing negative mental states, such as depression, anxiety and stress [21], due to the nature of their work. Unlike depression, burnout is specific to an individual’s relationship to his or her occupation and usually results from long-term exposure to occupational stress [22]. Burnout may lead to adverse outcomes, such as medical errors, suicide, depression and absenteeism [23,24]. It is known that stress and burnout are factors that can contribute to a decreased mental health [10,25]. The ICN [26] states that long and stressful shifts severely impact nurses’ mental health, resulting in nurses leaving or planning to leave the profession. Stress and burnout were recognized internationally as work hazards for nurses even before the pandemic [27]. Some argue that burnout in itself is a form of mental illness. However, a more common assumption has been that burnout causes mental dysfunction, such as anxiety and depression [28]. Temporal, physical, emotional and mental workloads, as well as job stressors such as time pressure, have in previous studies been positively associated with negative stress and burnout symptoms [28,29]. Previous studies have stated that nurses’ work environments contribute to high stress, job dissatisfaction and burnout [30,31]. Additionally, a previous study also stated that healthcare professionals (HCPs) in hospitals engage in many work-related tasks and experience relatively high levels of mental stress while caring for patients [32], while another study showed that workload and work pressure have an impact on job outcomes and quality of care [33]. Problems within the nurses’ working environments are described as concerns over inadequate staffing, ability to provide safe care, working long hours with high levels of fatigue and a sense of not being valued or involved in decision-making processes concerning patients [34,35,36]. Mental ill health has, in previous studies, been associated with different factors, such as long hours worked, work overload and work pressure, lack of control over work, lack of participation in decision making, poor social support as well as poor support from managers and an unclear work role [10,28,37]. Repeated exposure to stressful patient-related situations makes nurses especially susceptible to stress-related outcomes, such as emotional exhaustion and post-traumatic stress disorder (PTSD) [38]. Stress-related outcomes in nurses can lead to grave consequences, including depression, lower job satisfaction, increased risk of medical errors, lower productivity and higher turnover intentions [28]. Ignoring the signs of anxiety and depression presented by nursing professionals could increase physical and emotional stress for the individual but could also result in low quality patient care and higher work burden on the organizations [18].

However, even though there are investigations into nurses’ stress related to health and work environment, to our knowledge, there are no previous investigations on nurses´ total work situation, i.e., that investigates all areas of impact and determination for a sustainable working life, before and during the COVID-19 pandemic. It is, therefore, important to investigate what effect the COVID-19 pandemic had on healthcare organizations’ impact and determinant areas associated with nurses´ work-related mental health diagnoses.

The objective of this present study is to investigate the association between work-related mental-health diagnoses and different factors in nurses’ work situations before and during the COVID-19 pandemic. We want to test the hypothesis that there are no differences in nurses’ work related mental-health diagnoses in 2017 and 2020. The specific research questions are:Is there a difference between 2017 and 2020 regarding nurses’ work-related mental health?What associations are there between nurses’ work-related mental health and their work situation in 2017?What associations are there between nurses’ work-related mental health and their work situation in 2020, that is, during the COVID-19 pandemic?

This study also wants to increase knowledge and suggest measures against staff shortages and future challenges in healthcare.

## 2. Materials and Methods

### 2.1. Design

This longitudional study is part of a greater research programme, “Sustainable working life for all ages” [19,39]. In the spring of 2017, a baseline survey was performed where links to an online survey were sent out to all healthcare staff in the Swedish Region of Skane via their work e-mail. The follow-up study was performed in the fall of 2020, where all healthcare staff who were employed in 2017 and that were still employed in 2020 were invited to partake in the online survey.

### 2.2. Study Population

A link to a web survey was sent out via work e-mail to all employees in the Healthcare sector in the Swedish region of Skane, that is to all physiotherapists, occupational therapists, doctors, psychologists, nurses (including specialties such as midwife, CRNA, O.R nurse), nurse aids, etc. In total, the link to the 2017 survey was sent out to 22,935 employees, out of which 11,902 completed the survey. In this study, we will look specifically at nurses. In 2017 there were 9219 nurses (including specialist nurses) employed in the region, out of which 4692 completed the survey (50.9%). Some of the reasons for not answering the survey were wrong e-mail address, absence from work, lack of time and concerns about the manager finding out what they wrote. In 2020, data was collected via web-survey from the same study group during the second Covid 19-pandemic wave in Sweden, i.e., from September to December. Again, all healthcare staff who were employed in 2017 and that were still employed in 2020 were invited by e-mail to partake in the study. The survey link was sent out to all 18,143 staff, out of which 7781 participants responded. The number of nurses completing the web survey in 2020 was 3107 (40.1%). Some of the reasons for not responding to the survey were the same as in 2017, but there were also many nurses who stated that they wanted to prioritize their work and some nurses that had left the region.

Of the 4692 participants in the 2017 survey, the median age of the participants was 48 (23–67) and among the 3107 participants in 2020 it was 52 (26–70). In 2017, 90.4% who answered the questionnaire were women and in 2020 that number was 90.5%. A large majority (54.1% vs. 59.4% respectively in 2017 and 2020) of participants had worked as nurses for more than 16 years.

### 2.3. Themes in the Analysis Model

The theoretical SwAge-model (sustainable working life for all ages) [17,18,19] was used as the theme areas in the analysis with the intention of investigating factors concerning the complexity of the nurses’ work situations that could relate to their mental diagnoses caused by their work life and work environment. The SwAge-model consists of nine different impact and determinant areas that are important for a healthy and sustainable working life for all ages, and the four spheres of determination regarding employability. These four spheres and the nine impact and determinant areas are:I.The health effects of the work environment, which include the following areas of determination:
(1)Self-rated health, diagnoses and diverse physical and mental health functionality in work;(2)Physical work environment with unilateral movements, heavy lifting, risk of accidents, climate, chemical exposure and risk of contagion;(3)Mental work environment: stress and fatigue syndrome, threats and violence;(4)Working hours, work pace and possibility of recuperation during and between work shifts.II.Financial incentives are associated with society’s control of various financial motivations, such as through the social insurance system. Financial incentives include the following determinant area:
(5)The personal financial situation’s effects on individuals’ needs and willingness to work. Issues with employability due to ill health and lack of support can jeopardize inclusion in working life and cause an inferior financial situation for the individual, e.g., through sick leave, unemployment and early retirement.III.Relationships, social support and participation, i.e., attitudes in the social context in which the individual finds himself/herself, whether the individual feels included or excluded in the group and receives satisfactory social support from the environment when needed, which includes the following areas of determination:
(6)The effects of the personal social environment with family, friends and in the leisure context;(7)The social work environment with leadership, discrimination and the significance of the employment relationship context for individuals’ work.IV.Performance of duties and activities relating to individual and instrumental support, which includes the following areas of determination:
(8)Motivation, appreciation, satisfaction and stimulation when performing the work tasks;(9)Knowledge, skills, competence and competence development.

Seven out of the nine impact and determinate areas in the SwAge-model were used as independent variables in this investigation to analyze factors in the nurses’ work situation associated with work-related mental health diagnoses. The Health (1) area was used as the outcome/dependent variable in the analysis, i.e., mental health diagnoses caused by the work situation. However, the Personal finances (5) area was excluded since there were no data on the nurses’ private economic situation related to the work situation that could be used in the analysis.

### 2.4. Outcome Measures

The dependent variable was initially two individual statements: “I have a current diagnosis of exhaustion/stress” and “I have a current diagnosis of depression/anxiety”. These individual statements were put together into one variable regarding their work-related mental-health diagnoses as an outcome measure. The self-reported doctor’s diagnoses included in mental health were depression, anxiety, exhaustion and stress. The response options in the survey regarding diagnoses were taken from WHO’s ICD-10 codes.

The independent variables used in the univariate estimates and multivariate models were calculated using a categorical variable of diagnoses caused by work, i.e., mental health, as the outcome measures in association with the seven determinate areas for a healthy and sustainable worklife and employability in the SwAge-model that have been used in previous studies investigating factors associated to a sustainable working life (see above).

### 2.5. Questionns and Statments

The questionnaires were written in Swedish and contained 158 questions based on the SwAge-model that has been used in different investigations since 2004 [19]. However, in the questionnaire that was sent out in the second pandemic wave in 2020, additional questions about the COVID-19 situation were added. Some questions were simple yes/no questions and some were open answer questions where the participants could write freely. Most questions were designed as statements, a validated Lickert scale was used, and the participants had four answer options ranging from fully agree (1–2, i.e., Fully agree, partly agree) to fully disagree (3–4, i.e., partly disagree, fully disagree). The sample data was collected and handled by researcher KN. In the present study, 24 statements sorted into seven of the determinate areas of the SwAge-model were used. 

### 2.6. Statistical Analyses

Logistic regression analysis was used to test models to predict categorial outcomes and to assess how well a predictor variable associates with a categorial dependent variable [40]. The material was analyzed with the IBM SPSS software, version 27. Data are presented as odds ratios (ORs) with their 95% confidence intervals (CIs). Questions with four answer options were dichotomized for clear distinction of the participants’ experiences. A multivariate analysis of variance (MANOVA) was performed comparing and testing the statistical significance of the multivariate sample mean differences to see which statements in the multivariate model saw the most increase between 2017 and 2020. Mental health was the dependent variable and the 24 statements in the seven impact and determinate areas were the independent variables. As with the logistic regression analysis, the MANOVA was analyzed with the IBM SPSS software, version 27.

#### 2.6.1. Analyses within Each of the Seven Determinate Areas in the SwAge Analysis Model

Univariate logistic regression analysis was the first step to building multivariate models in each determinate area as well as for all determinate areas together. The univariate logistic regression analysis estimated for all statements within each of the seven determinate areas of the SwAge-model to investigate the association between the independent and the dependent variables. Initially, the associations for each statement were evaluated and the statements with *p*-values < 0.05, considered as the statistically significant level, were evaluated with other statements from the same determinate area. After this, the statements that continued to have a *p*-value < 0.05 were tested against the remaining statements one at a time. This continued for as long as the *p*-values for all included statements were <0.05.

#### 2.6.2. Analyses including the Seven Determinate Areas in the SwAge Analysis Model

After the initial univariate analysis, a modulation was made for each determinate area in the SwAge-model. All the selected statements from the seven included determinate areas of the SwAge-model were analyzed in a logistical regression model. Thereafter, the analysis moved to step 2, where the statistically significantly statements (*p*-values < 0.05) from each determinate area was added, one determinate area at a time. These statements were tested to form the final model. In step 3, the out-sorted statements from step 2 and from each of the seven determinate areas were added one at a time to test the robustness of the model [41]. The multivariate models were tested for collinearity.

### 2.7. Ethical Considerations

The study was performed in accordance with the Helsinki declaration [42] and Swedish laws [43]. The benefits the knowledge this study would generate was considered to outweigh the potential risks that the study could bring. Rules for the handling and storage of data was and will be followed in accordance with university policies as well as guidelines for handling sensitive data according to GDPR [44]. The study was approved by the Swedish Ethical Review Agency (number 2016/867 and 2020-01897).

## 3. Results

### 3.1. Findings

There was an increase in diagnosis for both examined areas, i.e., exhaustion/stress and depression/anxiety between 2017 and 2020 (see Table 1). Of note, 128 nurses that previously reported no mental health diagnosis stated that they had been diagnosed with exhaustion/stress and/or depression/anxiety in 2020. In the logistic regression analysis, these diagnoses were combined into one variable: “mental health diagnoses”.

### 3.2. Univariate Estimates and Multivariate Models for Work-Related Mental Health Diagnoses and Each of the Statements in the Analysed Areas

A logistic regression analysis was used to investigate which of the areas of importance for a healthy and sustainable work situation had the highest association with nurses’ work-related mental health diagnoses in 2017 and in the second wave of COVID-19 in 2020. The statements in each impact and determinate area were analyzed by area. There were seven impact and determinant areas included in the study, which were physical work environment (2); mental work environment (3); work pace, work time, recuperation (4); private social environment (6); work social environment, organization, leadership (7); motivation and satisfaction of and to work tasks (8); knowledge and competency (9).

In the impact and determinant area “physical work environment”, both included statements had a statistical association with nurses´ mental health diagnoses caused by their work for both 2017 and 2020 in the univariate estimates and in the multivariate model in 2017. However, the statement “For the most part I cannot cope with the physical work demands” also showed an association in the 2020 multivariate model.

All five statements in the impact and determinant area “Mental work environment” were statistically significant in the univariate estimates. In the 2017 multivariate model three statements were significant, which were “My work involves many psychologically heavy work tasks” (OR 1.78), “My work tasks usually clump together to the extent that I get frustrated” (OR 1.78) and “I wish for more opportunities to determine how to perform my work” (OR 1.60). In 2020, “My work tasks usually clump together to the extent that I get frustrated” (OR 2.05), “My work involves many psychologically heavy work tasks” (OR 1.76) and “I wish for greater control over my work (OR 1.75) showed significance.

All three of the investigated statements in the impact and determinant area “Work pace, work time, recuperation” showed an association in both the univariate estimates and the multivariate model for a healthy and sustainable working life. 

In the impact and determinant area “Private social environment”, both statements were statistically significant in the univariate estimates of 2017. The statement “I need to work more at home/care for relatives and will probably therefore work less in the future” showed an association in the multivariate model of 2017, whereas the statement “I want to spend more time enjoying leisure activities and will therefore work less in the future” showed an association in both the univariate estimates and multivariate model of 2020.

The area “Work social environment, organization, leadership” consisted of six statements that all indicated an association in the univariate estimates of 2017 and 2020. In 2017, three statements showed significance, which were “Not having enough staff means that I cannot perform my work in the way I want” (OR 2.04), “Big changes in my work situation causes me to want to leave” (OR 1.41) and “The social community at my workplace does not make me want to stay” (OR 1.33). In 2020, only two statements showed significance, including “I do not feel I have enough support from my co-workers” (OR 3.14) and “Big changes in my work situation causes me to want to leave” (OR 1.73).

Similar to the previous determinant area, all statements in the impact and determinate area “Motivation and satisfaction of and to work tasks” indicated an association in the univariate estimates of both 2017 and 2020. However, only two statements showed association in the 2017 and 2020 multivariate models: “I do not experience joy in my daily work” and “I do not experience satisfaction in my daily work”.

The impact and determinant area “Knowledge and Competency” consisted of two statements that were both found to be statistically significant in the univariate estimates of 2017 and 2020, but only the statement “I do not feel like my competencies are being utilized in a satisfactory way” showed association with nurses’ work-related mental health diagnoses in the 2017 and 2020 multivariate model (Table 2).

A multivariate analysis of variance (MANOVA) was performed in the multivariate model in order to see which of the 24 statements showed the most increase between 2017 and 2020. MANOVA was used since it does not affect the Type I error rate to the same extent as other independent tests. The results of the MANOVA mirrored the logistic regression analysis and the four statements that saw the most increase were “I do not have the time to perform the work duties I have planned for the day”, “I want to spend more time enjoying leisure activities and will therefore work less in the future”, “I do not feel I have enough support from co-workers” and “I do not experience joy in my daily work”. Results of the MANOVA showed that there was a statistical difference between the combined dependent variables. Wilks´Λ = 0.09, F(40,1582) = 4.200, *p* < 0.001, partial η^2^ = 0.096, observed power = 1.00. Based on the low Wilks´Λ, we want to be careful rejecting the null hypothesis. The observed power was 1.00, indicating that there was a 100% chance that the results could have been significant.

### 3.3. Multivariate Model of all Impact and Determinant Areas in the Work Situation in Association with Nurses’ Mental Health Diagnoses Caused by the Work Situation in 2017 and in 2020

In real life, nurses are not only affected by one of the impact and determinant areas from the SwAge-model, there is impact from all nine areas. Therefore, in the next step of the analysis we aimed to make a collected analysis of the seven relevant impact and determinate areas in this study. Hence, all the statements from the seven deteminant and impact areas for a sustainable healthy working life (the swAge-model) included in this investigation were modelled into a single multivariate model for each year, i.e., 2017 and for the second wave of COVID-19 in 2020. The variables that were statistically significant (*p*-value < 0.05) from each area were used in the modelling, and each of the eliminated statements (i.e., the variables not statistic significant in the earlier analysis of each area) were added once more one at a time to test the robustness of the model.

In the 2017 multivariate model, six statements showed significance: “I do not experience joy in my daily work” (OR 1.97), “My work involves many psychologically heavy work tasks” (OR 1.66), “The work pace in my daily work is too high” (OR 1.37), “I wish for more opportunities to determine how to perform my work” (OR 1.37), “My work tasks usually clump together to the extent that I get frustrated” (OR 1.34) and “I need to work more at home/care for relatives and will probably therefore work less in the future” (OR 1.27). In 2020, five statements showed significance: “I do not experience joy in my daily work” (OR 2.17), “I do not feel enough support from my co-workers” (OR 2.00), “My work tasks usually clump together to the extent that I get frustrated” (OR 1.81), “My work involves many psychologically heavy work tasks” (OR 1.69) and “I do not get enough rest/recuperation between work shifts (OR 1.41). There were no statistically significant statements in 2017 or in 2020 from the area “physical work environment” and “knowledge and competency” in the final total multivariate model, including all relevant impact and determinant areas for a healthy and sustaniable working life (Table 3).

### 3.4. Multivariate Model of the Work Situation in the Second Wave of COVID-19 in Association with Nurses’ Mental Diagnoses Caused by the Work Situation, including COVID-19-Specific Questions

COVID-19 had a significant impact on the healthcare systems in Sweden, particularly during the fall/winter of 2020. COVID-19-specific questions were, therefore, added to the investigation during the second wave of COVID-19 in 2020. Therefore, 25 COVID-19-specific variables in the seven impact and determinant areas were included in the next step of the analysis of the multivariate model regarding the second wave of COVID-19 in 2020 to see whether there were particular areas that affected the nurses. The statements in each impact and determinant area were analyzed within that particular determinant area. All statements showed significance in the univariate model. Sixteen statements remained significant in the multivariate model; out of these, five showed a slightly higher OR, which were “I do not feel enough support from my co-workers” (OR 2.86), “I do not experience joy in my daily work” (OR 2.46), “My workload has been higher during COVID-19 compared to my average workload” (OR 2.33), “My work tasks usually clump together to the extent that I get frustrated” (OR 2.07) and “I do not feel like my competencies are being utilized in a satisfactory way” (OR 2.02) (Table 4).

### 3.5. Final Multivariate Model with COVID-19-Specific Variables of the Work Situation in Association with Nurses’ Mental Health Diagnoses Caused by the Work Situation

We wanted to see which of the variables were most likely to impact nurses’ mental health and, therefore, be chosen for a final multivariate model. All statistically significant variables from Table 4 were added one at a time to form a multivariate model. The discarded statements were then added to the model one at a time to test the robustness of the model. In the end, the model consisted of eight statements that showed a connection with nurses’ mental health diagnoses (Table 5).

## 4. Discussion

Nurses are one of the biggest workgroups within the healthcare sector, and nursing is an important social security profession [7,8]. Unfortunately, many nurses are currently on short- or long-term sick leave, and too many nurses choose to leave the profession in the beginning of their educational training or a short time after their entry into the profession [10]. During the COVID-19 pandemic, nurses’ work situations were tested to the limit [1,2,3,4,5]. The aim of the study was, therefore, to investigate the association between work-related mental health diagnoses and nurses’ work situations in 2017 and 2020, i.e., before and during the second wave of the COVID-19 pandemic. With the intention of investigating the complexity of the nurses’ work situations, the swAge-model was used as the theme model in the analysis. In the results, we could see that the percentage of nurses having a diagnosis for exhaustion/stress had tripled between 2017 and 2020, and the percentage of nurses with a diagnosis of depression/anxiety had doubled. These are alarming numbers, and the fact that so many nurses suffer from work-related mental health issues is something that needs to be addressed and dealt with.

### 4.1. Impact and Determinant Areas Important for Nurses’ Mental Health Diagnoses Caused by Their Work

The seven different impact and determinant areas of importance for a healthy and sustainable working life were analyzed one at a time before multivariate modelling to understand the wider complexity of the nurses´ work situations in relation to mental health diagnoses caused by their work. Earlier studies stated the importance of investigating the total complexity of the work situation and not only one or two areas of importance for a sustainable working life if the intention is to develop practically important knowledge for measured activities [16,17,18,19,20,21]. The result of this investigation showed that all seven of the impact and determinant areas appeared to have an association with nurses’ mental health diagnoses caused by work. 

According to the result of the univariate estimates, the nurses felt unable to cope with the physical work demands in both 2017 and in 2020. A too demanding physical work environment is problematic for a sustainable working life [16,17,18]. A lot of nurses’ physical activity is spent standing and/or walking [45,46,47]; however, depending on where you work, the physical activity can also consist of working in strenuous work postures or moving patients from bed to wheelchair or on/off operating tables [48,49,50,51]. A physically demanding work environment could contribute to stress since people tend to get more tired from a physically demanding work environment, and if you are tired, you tend to not keep up with the work pace and be more sensitive to stressful situations, thus, increasing a vulnerability to mental health diagnoses, such as burnout [19,20].

The mental work environment was earlier described as a very important area for employees’ mental health [16,17,18,52]. In the nurses´ mental work environment area, all five statements were statistically significant in 2017 with “psychologically heavy work task” having a strong connection; in 2020, this statement was still significant but the “I wish for more opportunities to determine how to perform my work” statement showed a slightly higher association. Having a perceived sense of control is identified as important for the well-being and mental health of nurses [53]. Previous studies state that the more nurses are exposed to stressful situations, the more likely it is that it will drain their psychological resources and they will experience stress-related outcomes, i.e., their job demands exceed their job resources and the result can be poor mental health [54,55].

Rest and recuperation are important to the individual’s health and vital in a sustainable working life [16,17,18]. In the “work pace, work time, recuperation area” in the analysis, it was a “lack of time to perform work duties” that showed a high association in 2017 and it was still significant in the regression analysis in 2020; however, in 2020, it was the “accumulation of work tasks” that seemed to have a stronger association. Not having enough time for work tasks or feeling like the work tasks are piling up can cause frustration and ultimately lead to certain work tasks not being carried out and a wish to leave the profession [16,17,18]. A recent study found a strong negative association between high work time demands and emotional exhaustion [56]. A worst-case scenario is that lack of time can affect quality of care and/or affect nurses’ health [57,58]. A British survey [59] showed that, in some cases, this time constraint can result in malpractice and the neglect of patients. Recuperation between work shifts is important for all aspects of an individual’s well-being. Recovery is necessary for the body to reverse changes in the psychobiological system (such as increased heartrate from stressful work situations) [60].

In the impact and determinant area of “private social environment for a sustainable working life”, it was interesting that it was the “need to work more from home/care for relatives” that was statistically significant in 2017, whereas “wanting to spend more time enjoying leisure activities” showed as not being significant; however, in 2020, the roles were reversed. The balance between the work and the private social situation is important for employees’ sustainable working life [16,17,18]. Sweden, as a country, did not enforce lock-down during the COVID-19 pandemic. There were restrictions as to how and when you could go to gyms, for example, and a lot of activities were held on-line instead of in person. However, could the fact that there were restrictions in place influence the respondents’ feelings? A recent study showed that nurses felt it was important to leave all their experiences from working during the pandemic behind at work and when at home to focus on being at home cooking and cleaning as well as practicing self-care by exercising, walking, or spending time in nature [61].

Concerning the social environment at work, the organization and leadership are very important factors for a healthy and sustainable working life. In our study, the data from 2017 showed that lack of support from managers and co-workers seemed to have the least connection with nurses’ work-related mental health diagnoses. However, in 2020, lack of support from co-workers had one of the highest connections. Our study did not show any statistical significance regarding lack of support from managers in 2017 or in 2020. It is interesting that lack of support from co-workers changed from a low connection to having one of the highest connections with work-related mental health diagnoses. Is this due to the fact that nurses relied on support from colleagues more during the COVID-19 pandemic or has the pandemic simply put the spotlight on what was always there? Previous studies have shown that collegial support affects communication, organizational commitment, teamwork, stress, negative interaction, human relations, job satisfaction and the hierarchy in the workplace [16,17,18,62]. Positive social relations at work can ease the burden of emotional demands and work time demands [16,17,18,63].

Motivation and satisfaction regarding work tasks are important in order to have a healthy and sustainable working life. Our results show that in 2017, both lack of joy in the daily work as well as having no job satisfaction seemed to have high associations with mental health and work-related diagnoses. Lack of joy in the daily work continues to have increased association with work-related diagnoses both in the 2017 and the 2020 multivariate model. According to a study [64], experiencing joy at work is important both for the nurse and for healthcare in general. Several studies [65,66,67] show that job satisfaction is a vital component in nursing and that it is strongly related to factors such as job stress [16,17,18,67], intention to leave [16,17,18,65,67,68], quality of care [69] and patient satisfaction [67]. Studies have shown that nurses reported higher levels of job satisfaction when they felt high levels of support from their manager [17,70].

When it came to the impact and determinant area “knowledge and competency”, only the feeling of not having their competencies utilized showed to be statistically significant to nurses´ work-related mental health diagnoses. An earlier study [71] showed that good interaction between colleagues was a resource for high quality of care, which allowed everyone to use their competence well. Additionally, not being able to use their skills could most likely affect nurses’ willingness to stay in their workplace.

### 4.2. Multivariate Analysis of the Total Complexity in the Nurses’ Work Situation in Association with Work-Related Mental Diagnoses in 2017 and in 2020

In reality, each impact and determinant area is not operated separately. Therefore, all impact and determinant areas involved in this investigation were analyzed together in a total multivariate model to investigate the association between nurses’ work situations and nurses´ mental health diagnoses. Out of the original seven included impact and determinant areas, only five remained statistically significant and were, therefore, included in the final multivariate models for 2017 and 2020. The included areas were ”mental work environment”, ”work time, work pace, recuperation”, “private social environment”, “work social environment, organization, leadership” and “motivation and satisfaction of and to work tasks”. Only three statements showed an association with nurses’ work-related mental health diagnoses in both 2017 and 2020, they were “My work involves many psychologically heavy work tasks”, “My work tasks usually clump together to the extent that I get frustrated” and “I do not experience joy in my daily work”. Two additional statements showed significance in the 2020 model, which were “I do not feel that I get enough rest/recuperation between work shifts” and “I do not feel enough support from my co-workers”. Feeling that you are unable to provide proper care to patients can lead to ethical and moral stress among nurses, which, in turn, can affect their health and psychological well-being [72,73] and cause job dissatisfaction [73]. Nurses and other healthcare workers’ mental health diagnoses have been shown to threaten the quality of care and patient safety [74,75,76]; this adds further importance to the fact that healthcare organizations must take the nurses’ work situation very seriously.

### 4.3. The COVID-19 Pandemics Effect on Nurses’ Work Situation

Several studies have shown that many healthcare workers have experienced anxiety, depression [77,78,79] and burnout [79] during the COVID-19 pandemic. The final multivariate model showed that anxiety over being seriously ill can be associated with nurses’ work-related mental health diagnosis. The COVID-19 pandemic had a huge impact on the healthcare organizations, with many millions of people, including nurses, becoming infected by the virus, thus, causing an increased workload for nurses [1,2,3,4,5]. Being at risk of being infected by COVID-19, becoming seriously ill, dying or infecting others has been cited as a major risk for work-related mental illness for healthcare workers during the COVID-19 pandemic. In this investigation, some of the variables have not shown a change between 2017 and 2020, most likely indicating that the COVID-19 pandemic did not impact these particular variables. The result in the final multivariate model did show that the nurses‘ increased risk of being infected by COVID-19 in their workplace was associated with work-related mental health illness. However, issues related to personal protective equipment was not statistically significant. Instead, the result showed that especially lack of support from co-workers, lack of joy in their daily work as well as an increased workload and the accumulation of work tasks showed increased associations with nurses’ work-related mental health diagnoses during the COVID-19 pandemic. A previous systematic review stated that the COVID-19 pandemic forced nurses to have a greater workload, but also that many nurses had trouble falling asleep and/or not getting enough sleep, which they attributed to lack of time to decompress mentally between work shifts [10]. Additionally, nurses felt like their competencies were not utilized in a satisfactory way. Not being given the opportunity to use their skills or feeling that the organization does not utilize or appreciate the skills and knowledge that the employees possess has, in previous studies, been associated with a lack of job satisfaction and motivation and could lead to employees not wanting to continue working at the workplace [16,17,18].

The pandemic put the spotlight on nurses’ work situations, but is the spotlight pointing in the right place? Many healthcare organizations had problems in their work environment prior to the pandemic (including lack of staff and the work situation). Have certain aspects of the nurses’ work situation become more important during the COVID-19 pandemic or has the pandemic simply shown cracks in the façade? Perhaps only the future can tell since we are still living with the pandemic.

### 4.4. Limitations

One limitation of the study is that we had a large percentage of non-responders, the answer rate was 50.1% and 40.1% in 2017 and 2020, respectively. However, considering that it was a survey, the low response rate was expected, and we are very grateful to those nurses who took the time and answered the survey, especially in 2020 when there was an on-going pandemic. Another limitation is that when you use dichotomization, there is always a risk of losing valuable information. The dichotomization was made by an experienced researcher who thoroughly made considerations in which response choice dichotomization was used. The fifth impact and determinate area, i.e., "personal finances” was not included in the study since there was no data on the nurses’ private economic situation related to the work situation that could be used in the analysis. However, this area could have an impact on the results, for instance, if nurses went to work despite being sick due to not being able to afford the loss in pay. This, in turn, could contribute an added stress. In this study, we have used the respondents’ self-reported doctors’ diagnoses that they felt were caused by their work. One opportunity could have been to use registers with reported work-related illnesses. However, in these registers there are only those diagnoses that have been deemed as work injuries and, therefore, the diagnoses in our study would probably not have been reported. It is also well known that the number of reported work-related illnesses is underreported [80]. Therefore, we found it more valuable to ask the nurses about which of their diagnoses they felt had been caused by their work. One limitation is the possibility of responders misunderstanding the questions regarding their current mental health diagnoses since no specific timeframe was given, i.e., “I was diagnosed with exhaustion/stress or depression/anxiety within the last six months”. Another limitation is the low score on Wilks´Λ, which would mean that we cannot rule out the possibility of other factors influencing nurses’ work-related mental health other than those we have presented.

## 5. Conclusions

Based on the results of this survey, there were some differences in what was associated with the nurses’ work-related mental health diagnoses in 2017 and in 2020. The COVID-19 pandemic put nurses’ working situations to a severe test. The result from this comparative analysis, where we examined the work situations and work-related mental health diagnoses before and during the second wave of the COVID-19 pandemic, showed that increased workload and experiencing a lack of joy in the nurses´ daily work as well as experiencing a perceived lack support from their co-workers had the strongest association with nurses’ work-related mental health diagnoses in 2020. It is hard to get around the fact that nurses will continue to face psychologically and physically heavy work tasks, but it is important for organizations to have an open climate so that nurses can talk about their experiences. For nurses to have more opportunities to determine how to perform their work tasks, it is important that they feel like they have a safe work environment and that they have adequate staff and resources and feel involved in decision-making. This study’s analysis model is based on theories about factors that influence a healthy and sustainable working life, and the results are consistent with what the SwAge-model has previously shown [16,17,18].

Nurses and other healthcare workers’ mental health diagnoses have been shown to threaten the quality of care and patient safety [74,75]. Therefore, the result from this study investigating nurses’ work-related mental health diagnoses could be important knowledge for the future development of healthcare organizations. The results from this study could also be used by hospitals and ministries of health, etc. as a template to improve the working conditions and quality of life at work for nurses. If these two things improve, perhaps nurses would be more inclined to remain in their current workplace/profession. Future research regarding the long-term impact from COVID-19 on all areas of nurses’ professional and personal lives is needed.

## Figures and Tables

**Table 1 ijerph-19-15569-t001:** Percentage of nurses diagnosed with exhaustion/stress and/or depression/anxiety in 2017 and 2020.

Diagnosis	2017	2020
Exhaustion/stress	8.1%	26.4%
Depression/anxiety	5.3%	10.2%

**Table 2 ijerph-19-15569-t002:** Univariate and multivariate variables 2017 and 2020. Univariate estimates and multivariate models in each of the analyzed areas between the statements (agree vs. disagree) and work-related mental health diagnoses and other factors. OR = Odds ratio; CI = Confidence interval. * The variable shows no statistical significance in the multivariate modelling and is, therefore, not included in the final multivariate model shown in this column.

		Univariate Estimates for Each Variable in 2017		Multivariate Model in Each Area in 2017		Univariate Estimates for Each Variable in 2020		Multivariate Model in Each Area in 2020	
Area	Statement	OR	CI 95%	OR	CI 95%	OR	CI 95%	OR	CI 95%
Physical work environment	For the most part I cannot cope with the physical work demands	1.74	1.22–2.48	1.67	1.17–2.40	2.01	1.17–3.46	2.01	1.17–3.46
	My current work is too physically straining for my health	1.37	1.08–1.73	1.32	1.04–1.68	1.44	0.99–2.08	*	*
Mental work environment	My work involves many psychologically heavy work tasks	2.21	1.75–2.78	1.78	1.40–2.26	2.17	1.60–2.95	1.76	1.28–2.41
	I wish for more opportunities to determine how to perform my work	2.00	1.63–2.45	1.60	1.29–1.98	1.81	1.37–2.40	*	*
	I wish for greater control over my work	1.89	1.55–2.31	*	*	2.24	1.69–2.97	1.75	1.30–2.36
	At my workplace there are not enough possibilities to be re-allocated to less demanding work tasks for those who need it	1.36	1.10–1.68	*	*	1.29	0.96–1.72	*	*
	My work tasks usually clump together to the extent that I get frustrated	2.32	1.89–2.84	1.78	1.45–2.23	2.63	1.98–3.49	2.05	1.51–2.78
Work pace, work time, recuperation	I do not feel like I get enough rest/recuperation between work shifts	1.83	1.50–2.23	1.44	1.17–1.78	2.23	1.68–2.97	1.73	1.27–2.35
	I do not have time to perform the work duties I have planned for the day	1.74	1.42–2.14	1.312	1.06–1.63	2.50	1.84–3.39	1.79	1.27–2.53
	The work pace in my daily work is too high	2.27	1.85–2.78	1.88	1.51–2.34	2.22	1.67–2.96	1.55	1.12–2.15
Private social environment	I want to spend more time enjoying leisure activities and will therefore work less in the future	1.33	1.02–1.74	*	*	1.51	1.04–2.19	1.51	1.04–2.19
	I need to work more at home/care for relatives, and will probably therefore work less in the future	1.46	1.17–1.81	1.42	1.14–1.77	1.02	0.72–1.43	*	*
Worksocial environment, organization, leadership	The social community at my workplace does not make me want to stay	1.50	1.20–1.88	1.33	1.06–1.68	1.47	1.06–2.04	*	*
	Big changes in my work situation causes me to want to leave	1.81	1.45–2.26	1.41	1.12–1.78	2.15	1.53–3.02	1.73	1.21–2.47
	I do not feel I have enough support from my closest manager	1.24	1.01–1.52	*	*	1.56	1.16–2.08	*	*
	I do not feel I have enough support from my co-workers	1.36	1.00–1.84	*	*	3.53	2.41–5.17	3.14	2.11–4.67
	I feel bullied or shut out from the community at my work place	1.91	1.21–3.02	*	*	2.35	1.26–4.39	*	*
	Not having enough staff mean that I cannot perform my work in the way I want	2.26	1.83–2.79	2.04	1.64–2.54	1.44	1.08–1.91	*	*
Motivation and satisfaction of and to work tasks	I do not feel like my daily work is meaningful	2.22	1.63–3.01	*	*	2.35	1.47–3.76	*	*
	I do not feel like my work is stimulating	2.14	1.67–2.73	*	*	1.99	1.36–2.90	*	*
	I do not experience joy in my daily work	2.73	2.20–3.39	2.02	1.45–2.82	3.45	2.53–4.70	2.39	1.48–3.84
	I do not experience satisfaction in my daily work	2.48	2.01–3.08	1.49	1.07–2.07	3.10	2.26–4.24	1.64	1.01–2.66
Knowledge and Competency	I do not get enough opportunities at work to utilise my skills and knowledge	1.74	1.36–2.23	*	*	1.70	1.17–2.46	*	*
	I do not feel like my competencies are being utilised in a satisfactory way	1.82	1.46–2.28	1.64	1.21–2.22	2.10	1.53–2.90	2.17	1.42–3.32

**Table 3 ijerph-19-15569-t003:** The final multivariate model for all areas and statement together for 2017 and for 2020. Statistically significant variables in relation to nurses’ work-related mental health diagnosis in 2017 and 2020. OR = Odds ratio; CI = Confidence intervl. Nagelkerke R square 0.073. * The variable shows no statistical significance in the multivariate modelling and is, therefore, not included in the final multivariate model shown in this column.

		2017		2020	
Area	Statement	OR	CI 95%	OR	CI 95%
Mental work environment	My work involves many psychologically heavy work tasks	1.66	1.29–2.13	1.69	1.22–2.34
	I wish for more opportunities to determine how to perform my work	1.37	1.10–1.72	*	*
	My work tasks usually clump together to the extent that I get frustrated	1.34	1.06–1.71	1.81	1.33–2.48
Work time, work pace, recuperation	I do not feel that I get enough rest/recuperation between work shifts	*	*	1.41	1.03–1.93
	The work pace in my daily work is too high	1.37	1.07–1.74	*	*
Private social environment	I need to work more at home/care for relatives, and will probably therefore work less in the future	1.27	1.02–1.56	*	*
Work social environment, organization, leadership	I do not feel enough support from my co-workers	*	*	2.00	1.31–3.08
Motivation and satisfaction of and to work tasks	I do not experience joy in my daily work	1.97	1.56–2.48	2.17	1.52–3.09

**Table 4 ijerph-19-15569-t004:** Univariate estimates and the total multivariate model including all seven investigated areas in 2020 with COVID-19-specific questions. Statistically significant variables in relation to work-related mental health diagnoses in 2020. OR = Odds ratio; CI = Confidence interval. Nagelkerke R square 0.115. * The variable shows no statistical significance in the multivariate modelling and is, therefore, not included in the final multivariate model shown in this column.

		Univariate Estimates		Multivariate Model	
Area	Statement	OR	CI 95 %	OR	CI 95%
Physical work environment	For the most part I cannot cope with the physical work demands	2.01	1.17–3.46	1.93	1.10–3.38
	My current work is too physically straining for my health	1.44	1.00–2.08	*	*
	The hygiene routines in my daily work are not enough to protect me from serious risk of being infected by COVID-19	1.50	1.05–2.15	*	*
	I do not experience the personal protective equipment (PPE) as satisfactory from an infection protection point of view	1.48	1.03–2.12	*	*
	The accessibility to proper PPE has not been enough to perform my work duties safely	1.17	0.83–1.65	*	*
	The PPE is designed in a way that makes it difficult to perform my work duties safely	1.12	0.77–1.61	*	*
	In my daily work there are obstacles that prevent employees from fully compling with COVID-19 safety procedures	1.22	0.90–1.65	*	*
	The PPE prevents me from performing my work duties in a (for me) comfortable and satisfactory way	1.18	0.89–1.57	*	*
	The measures for preventing ill health and disease among the staff during the COVID-19 pandemic are not good enough at my workplace	1.22	0.88–1.68	*	*
	I feel that there have been significant risks of being infected by COVID-19 in my workplace	1.51	1.13–2.01	1.50	1.13–2.00
	My work situation during COVID-19 has not contained more physical load when compared to normal circumstances	1.13	0.81–1.58	*	*
Mental work environment	My work tasks usually clump together to the extent that I get frustrated	2.63	1.98–3.49	2.07	1.51–2.83
	My work involves many psychologically heavy work tasks	2.17	1.60–2.95	1.81	1.31–2.51
	I wish for more opportunities to determine how to perform my work	1.81	1.37–2.40	*	*
	I wish for greater control over my work	2.24	1.69–2.97	1.73	1.18–2.54
	My work situation during COVID-19 has been more stressful in comparison to normal circumstances	1.63	1.22–2.18	*	*
	I have had anxiety over myself being severely ill with COVID-19	1.53	1.16–2.03	*	*
	I have had anxiety over dying due to COVID-19	1.44	1.06–1.97	*	*
Work time, work pace, recuperation	My workload has been higher during COVID-19 compared to my average workload	1.38	1.04–1.83	2.33	1.19–4.56
	I do not have time to perform the work duties I have planned for the day	2.50	1.84–3.39	1.70	1.20–2.42
	I do not feel that I get enough rest/recuperation between work shifts	2.23	1.68–2.97	1.68	1.22–2.30
	The work pace in my daily work is too high	2.22	1.67–2.96	1.56	1.13–2.17
	My work situation during COVID-19 has had a negative impact on my ability to recuperate during work shifts due to reduced possibilities to take breaks, etc.	1.60	1.18–2.16	*	*
	My work situation during COVID-19 has had a negative impact on my ability to recuperate between work shifts	1.86	1.38–2.52	*	*
	I have not been able to take my vacation the way I had planned due to COVID-19	1.21	0.77–1.90	*	*
Private social environment	I want to spend more time enjoying leisure activities and will therefore work less in the future	1.51	1.04–2.19	*	*
	I need to work more at home/care for relatives, and will probably therefore work less in the future	1.02	0.72–1.43	*	*
	I feel that I have risked getting infected by COVID-19 in my leisure time (in the store, trip to/from work, etc.)	1.32	0.99–1.75	*	*
	My work situation during COVID-19 has had a negative impact on my private life (my family, partner, etc.)	1.70	1.28–2.25	1.42	1.05–1.92
	I have felt concern about a close relative being or getting severely ill by COVID-19	1.74	1.29–2.36	*	*
	I am/have been concerned that I will bring the COVID-19 virus home from work, which will infect family/friends, etc.	1.72	1.29–2.28	*	*
	I am/have been concerned that I will bring the COVID-19 virus from my private life and infect people and risk groups at my work	1.41	1.06–1.88	*	*
Work social environment, organization, leadership	The social community at my workplace does not make me want to stay	1.47	1.06–2.04	*	*
	Big changes in my work situation causes me to want to leave	2.15	1.53–3.02	1.64	1.11–2.42
	I do not feel enough support from my co-workers	3.53	2.41–5.17	2.86	1.84–4.44
	I do not feel I have enough support from my closest manager	1.56	1.16–2.08	*	*
	I feel bullied or shut out from the community at my workplace	2.35	1.26–4.39	*	*
	Not having enough staff means that I cannot perform my work in the way I want	1.44	1.08–1.91	*	*
	My closest manager has not given me enough support during the COVID-19 pandemic	1.62	1.20–2.18	*	*
	I have not received enough information/knowledge from management to perform my work duties in a satisfactory way during the COVID-19 pandemic	1.35	0.96–1.89	*	*
Motivation and satisfaction of and to work tasks	I do not experience joy in my daily work	3.45	2.53–4.70	2.46	1.50–4.04
	I do not feel like my daily work is meaningful	2.35	1.47–3.76	*	*
	I do not feel like my work is stimulating	1.99	1.36–2.90	*	*
	I do not experience satisfaction in my daily work	3.10	2.26–4.24	1.69	1.03–2.78
	The COVID-19 pandemic has not increased my motivation for my work tasks	1.43	0.98–2.10	*	*
	At my workplace there are not enough possibilities to be re-allocated to less demanding work tasks for those who need it	1.29	0.97–1.72	*	*
Knowledge and Competency	I do not get enough opportunities at work to utilise my skills and knowledge	1.70	1.17–2.46	*	*
	I do not feel like my competencies are being utilised in a satisfactory way	2.10	1.53–2.90	2.02	1.31–3.10
	I have not received enough information, knowledge, and competence development at work in order to feel safe performing my work tasks during the COVID-19 pandemic	1.65	1.22–2.24	1.49	1.09–2.03

**Table 5 ijerph-19-15569-t005:** Final multivariate model with COVID-19-specific variables. Statistically significant variables in relation to work-related mental health diagnoses in 2020. OR = Odds ratio; CI = Confidence interval. Cox and Snell R Square 0.058; Nagelkerke R square 0.121.

		Multivariate Model	
Area	Statement	OR	CI 95%
Mental work environment	I wish for greater control over my work	1.45	1.04–2.01
	My work involves many psychologically heavy work tasks	1.72	1.23–2.40
	I have had anxiety over myself being severely ill with COVID-19	1.40	1.03–1.89
	My work tasks usually clump together to the extent that I get frustrated	1.91	1.36–2.68
Work time, work pace, recuperation	I do not feel that I get enough rest/recuperation between work shifts	1.41	1.01–1.95
Work social environment, organization, and leadership	I do not feel enough support from my co-workers	1.96	1.27–3.01
	Not having enough staff means that I cannot perform my work in the way I want	1.53	1.08–2.16
Motivation and satisfaction of and to work tasks	I do not experience joy in my daily work	2.14	1.49–3.09

## Data Availability

Not applicable.

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
