# Peer review of "Nurses’ Work-Related Mental Health in 2017 and 2020—A Comparative Follow-Up Study before and during the COVID-19 Pandemic"

_ijerph, 2022, doi:10.3390/ijerph192315569_

Round 1
Reviewer 1 Report
Nurses’ Work-Related Mental Health in 2017 and 2020: A Comparative Follow-Up Study Before and During the Covid-19 Pandemic
International Journal of Environmental Research and Public Health
The data summarized in this manuscript have promise, however are incomprehensible in their current form. The amount of data analyses are vast, yet there is no overarching story that is told with them.
One of the most glaring omissions is the percentage of participants who either had a diagnosis of exhaustion/stress or depression/anxiety in 2017 compared to 2020.
The study is purported to be longitudinal in design, implying that participants’ responses were matched between 2017 and 2020. If so, it would be possible to compute difference scores on each of the 24 predictor statements (using the original 4-level scale) and determine
1. Which statements, of the 24 predictors, saw the most increase within that period? Did the overall difference change within the time period? This could be done with a MANOVA of the difference scores.
2. Which statements’ difference scores predict differences in the mental health diagnoses over time?
If the data were not matched to participants, the data could still be made more comprehensible and meaningful:
1. Both mental health outcome variables should be analyzed separately and then predicted using logistic regression for each of the 7 areas, keeping the responses for each statement on a 1 (fully agree) to 4 (fully disagree) scale. This should be done for both the 2017 and 2020 data.
2. Alternatively, the mental health outcome variables could be combined but recoded into three, rather than two, categories: 0 = no self-reported mental health issue on either statement, 1 = “yes” on one statement, 2 = “yes” on both statements. In its current form, having either one or two diagnoses is treated equally, reducing both variability and the value of prediction. This type of treatment of the outcome variable would then lend itself to being predicted using ordinal regression.
Stating which odds ratio is the highest among those that are very similar in value is not a practical way to determine which statement best reflects the risk of a mental health diagnosis. The vast majority of the CIs of the odds ratios overlap with one another both within areas and across comparison years (Table 3), indicating no significant difference between odds ratios; the differences are within the scope of normal error variance.
Other areas of concern:
Not sure what the demographic information in Table 1 adds. For instance, it makes sense that if this is a longitudinal study with the same participants at Wave 1 and Wave 2, the sample itself would be roughly 3 years older. There are no appreciable differences between Waves 1 and 2 for the rest of the information provided. It seems better and more efficient to simply state the mean age of the sample at Wave 1, percentage female, etc. in the text. Given that the majority of the demographic variables are not used in any way in subsequent analyses, they add very little to the overall picture.
Table 2 lists the percentage of respondents in 2017 and in 2020 who responded “agree” on 24 statements. It is quite noticeable that most of those statements showed lower percentages of participants agreeing in 2020 than in 2017, which is counter-intuitive and needs to be addressed.
Introduction
The Introduction section could be better organized. In its current form, it reads like a laundry list of studies that are not tightly connected.
Author Response
Response to Reviewer 1
Dear Reviewer 1,
Thank you so much for taking the time to read our manuscript. Your comments were very helpful.
- “The data summarized in this manuscript have promise, however are incomprehensible in their current form. The amount of data analyses are vast, yet there is no overarching story that is told with them”. Reply: In order to achieve an overarching story we have worked on the text between the "tables"
- “One of the most glaring omissions is the percentage of participants who either had a diagnosis of exhaustion/stress or depression/anxiety in 2017 compared to 2020.” Reply: We have added percentage of the nurses that had a diagnosis of exhaustion/ stress and/or depression/ anxiety.
- “The study is purported to be longitudinal in design, implying that participants’ responses were matched between 2017 and 2020. If so, it would be possible to compute difference scores on each of the 24 predictor statements (using the original 4-level scale) and determine Which statements, of the 24 predictors, saw the most increase within that period? Did the overall difference change within the time period? This could be done with a MANOVA of the difference scores. Which statements’ difference scores predict differences in the mental health diagnoses over time?” Reply: In this article we have not looked at predictors, we have another article coming up that will focus on predictors. We have made a MANOVA as suggested, and the results pretty much mirrored the logistic regression analysis. This is addressed in the text.
- “If the data were not matched to participants, the data could still be made more comprehensible and meaningful:
- Both mental health outcome variables should be analyzed separately and then predicted using logistic regression for each of the 7 areas, keeping the responses for each statement on a 1 (fully agree) to 4 (fully disagree) scale. This should be done for both the 2017 and 2020 data.
- Alternatively, the mental health outcome variables could be combined but recoded into three, rather than two, categories: 0 = no self-reported mental health issue on either statement, 1 = “yes” on one statement, 2 = “yes” on both statements. In its current form, having either one or two diagnoses is treated equally, reducing both variability and the value of prediction. This type of treatment of the outcome variable would then lend itself to being predicted using ordinal regression.”
REPLY: The data is matched to the participants
- “Stating which odds ratio is the highest among those that are very similar in value is not a practical way to determine which statement best reflects the risk of a mental health diagnosis. The vast majority of the CIs of the odds ratios overlap with one another both within areas and across comparison years (Table 3), indicating no significant difference between odds ratios; the differences are within the scope of normal error variance.” Reply: Some variables have not changed between 2017 and 2020 most likely indicating that Covid-19 did not impact these variables. This is addressed in the text.
- “Other areas of concern: Not sure what the demographic information in Table 1 adds. For instance, it makes sense that if this is a longitudinal study with the same participants at Wave 1 and Wave 2, the sample itself would be roughly 3 years older. There are no appreciable differences between Waves 1 and 2 for the rest of the information provided. It seems better and more efficient to simply state the mean age of the sample at Wave 1, percentage female, in the text. Given that the majority of the demographic variables are not used in any way in subsequent analyses, they add very little to the overall picture.” Reply: Table 1 was there for clarity; we know that it does not add anything, but we just wanted to show the demographic information. But we have removed this and kept the data in the text instead as per your suggestion.
- “The Introduction section could be better organized. In its current form, it reads like a laundry list of studies that are not tightly connected.” Reply: We have tried to organize the introduction in a better way.
Kind regards!
Reviewer 2 Report
This study adresses an important issue for health systems , the mental health of health worker, the nurses. However , in the conclusion it would be interesting to discuss how the study can be use by hospital , ministry of health etc. to improve the working conditions and quality of life at work of these fundamental workers for the heath sytems ?
Author Response
Response to Reviewer 2
Dear Reviewer 2,
Thank you so much for taking the time to read our manuscript. Your comments were very helpful.
“This study adresses an important issue for health systems , the mental health of health worker, the nurses. However , in the conclusion it would be interesting to discuss how the study can be use by hospital , ministry of health etc. to improve the working conditions and quality of life at work of these fundamental workers for the heath sytems ?”
Reply: As per your suggestion we have added the discussion regarding the use of this study in the conclusion.
Kind Regards!
Round 2
Reviewer 1 Report
Nurses’ Work-Related Mental Health in 2017 and 2020: A Comparative Follow-Up Study Before and During the Covid-19 Pandemic - Revision
International Journal of Environmental Research and Public Health
Results
I was happy to see the inclusion of a basic comparison of nurses’ mental health diagnoses between 2017 and 2020, which strongly reflected what the authors wish us to know: the year of Covid was very difficult for nurses, as one diagnosis tripled (exhaustion/stress) and the other doubled (depression/anxiety) within that time period.
Something I would like to see in the description of the outcome measures (Section 2.4) is a clarification that the questions pertained to current diagnoses rather than past diagnoses. Was there a time period given in the two statements (“I have a diagnosis of exhaustion/stress” “I have a diagnosis of depression/anxiety”) – that is, were participants given a timeframe (such as within the last six months)? I would certainly hope that this was made clear to participants so that the increase over the time period is not an artifact of those having diagnoses in 2017 continuing to indicate having the diagnosis if it was not current. If no timeframe was provided, the authors need to mention that as a limitation.
I still find the myriad of so many statistics contained in the tables, then repeated in the text, to be dizzying. The story the authors wish to tell gets very muddied in the process. I strongly urge some streamlining of the results and discussion.
First, odds ratios with overlapping confidence intervals are not significantly different from one another; however, a good deal of the text compares these odds ratios with language stating that one ratio was the “highest” of the group, implying statistically different ratios. This is akin to stating one mean is higher than another when they are not statistically different. My suggestion is to simply keep the information in Table 2 but eliminate the text. At the very least, the implying of statistical differences (i.e., “the highest OR…”) needs to be changed.
Table 3: I strongly suggest omitting the reporting of ORs in Table 3 and replacing that information with the statistics of the multivariate logistic regressions for 2017 and 2020, as described in Section 3.2. This would be more informative than reporting the ORs for each statement/year. That way, reader would know not only which statements significantly predicted a diagnosis for each year, but would be able to peruse the actual statistics and p-values associated with each statement.
I repeat that suggestion for reporting the information that is currently in Table 4 regarding the Covid-specific questions. Keep the ORs associated with each statement, but then include a Table 5 that reports the statistics associated with the logistical regression for each statement included in the multivariate regression. This information would be much more informative than merely reporting which statements significantly predicted a diagnosis.
Other
Table 2 shows five statements into the Mental Work Environment area and three in the Work pace, work time, recuperation area but four statements within each area are referred to in the text (Lines 285 and 291). It appears that the statement “My work tasks usually clump together to that extent that I get frustrated” was mistakenly categorized in the former, rather than the latter, area.
Author Response
Response to Reviewer 1
Thank you again for taking time out of your life to read through our manuscript and offer input and suggestions, we are confident that your insights have made our manuscript better. We have responded to your input below (in red).
“I was happy to see the inclusion of a basic comparison of nurses’ mental health diagnoses between 2017 and 2020, which strongly reflected what the authors wish us to know: the year of Covid was very difficult for nurses, as one diagnosis tripled (exhaustion/stress) and the other doubled (depression/anxiety) within that time period”.
Thank you for bringing this to our attention so that we could add this information. We are happy that this is clearer now.
“Something I would like to see in the description of the outcome measures (Section 2.4) is a clarification that the questions pertained to current diagnoses rather than past diagnoses. Was there a time period given in the two statements (“I have a diagnosis of exhaustion/stress” “I have a diagnosis of depression/anxiety”) – that is, were participants given a timeframe (such as within the last six months)? I would certainly hope that this was made clear to participants so that the increase over the time period is not an artifact of those having diagnoses in 2017 continuing to indicate having the diagnosis if it was not current. If no timeframe was provided, the authors need to mention that as a limitation”.
Thank you for this important input. The study was originally meant to be a cross-sectional study, when the second survey went out the wording in the questions were unfortunately not changed. We have addressed this in Section 2.4 and in limitations.
“I still find the myriad of so many statistics contained in the tables, then repeated in the text, to be dizzying. The story the authors wish to tell gets very muddied in the process. I strongly urge some streamlining of the results and discussion”. Thank you for pointing this out.
We have revised the text in the results and discussion section as per your suggestion by adding extra explanatory texts in the tables and by removing unnecessary statistics in the text. All changes are marked with yellow.
“First, odds ratios with overlapping confidence intervals are not significantly different from one another; however, a good deal of the text compares these odds ratios with language stating that one ratio was the “highest” of the group, implying statistically different ratios. This is akin to stating one mean is higher than another when they are not statistically different. My suggestion is to simply keep the information in Table 2 but eliminate the text. At the very least, the implying of statistical differences (i.e., “the highest OR…”) needs to be changed”.
We have revised the text as per your last suggestion, that is changed the implication of statistical difference. However, we feel it is important to keep the text since we want to reach both qualitative and quantitative researchers. Some readers of scientific texts prefer to read the results by looking at the tables, whilst others prefer reading the results in the text. In order to make the results more accessible for readers who are not so well versed in statistical methods we have chosen to present the text in both tables as well as text. This also gives us a chance to highlight the most important findings in the text.
“Table 3: I strongly suggest omitting the reporting of ORs in Table 3 and replacing that information with the statistics of the multivariate logistic regressions for 2017 and 2020, as described in Section 3.2. This would be more informative than reporting the ORs for each statement/year. That way, reader would know not only which statements significantly predicted a diagnosis for each year, but would be able to peruse the actual statistics and p-values associated with each statement”.
We have revised the text and added further clarifying comments in Table 3. Table 3 is a continuation from Table 2. Since the data in Table 3 is a multivariate model the statistically significant variables strengthen each other.
“I repeat that suggestion for reporting the information that is currently in Table 4 regarding the Covid-specific questions. Keep the ORs associated with each statement, but then include a Table 5 that reports the statistics associated with the logistical regression for each statement included in the multivariate regression. This information would be much more informative than merely reporting which statements significantly predicted a diagnosis”.
We have added clarifying comments to Table 4. Variables that did not show any statistical significance at the multivariate modelling was not included in the final multivariate model shown in this table. We have added Table 5 where the final multivariate model with relevant Covid-19 specific variables are presented.
Other
“Table 2 shows five statements into the Mental Work Environment area and three in the Work pace, work time, recuperation area but four statements within each area are referred to in the text (Lines 285 and 291). It appears that the statement “My work tasks usually clump together to that extent that I get frustrated” was mistakenly categorized in the former, rather than the latter, area”.
Thank you for bringing this to our attention, we have revised this in the text. The analyses are correct but unfortunately the text was incorrect. The statement “My work tasks usually clump together to that extent that I get frustrated” belongs in the Mental Work Environment area.
Sincerely
The authors
